# Effect of Different Coffee Brews on Tryptophan Metabolite-Induced Cytotoxicity in HT-29 Human Colon Cancer Cells

**DOI:** 10.3390/antiox11122458

**Published:** 2022-12-14

**Authors:** Luigi Castaldo, Marianna Toriello, Luana Izzo, Raffaele Sessa, Sonia Lombardi, Silvia Trombetti, Yelko Rodríguez-Carrasco, Alberto Ritieni, Michela Grosso

**Affiliations:** 1Department of Pharmacy, University of Naples “Federico II”, 49 via Domenico Montesano, 80131 Napoli, Italy; 2Laboratory of Biochemistry and Molecular Biology, Department of Molecular Medicine and Medical Biotechnology, School of Medicine, University of Naples Federico II, 80138 Napoli, Italy; 3Laboratory of Food Chemistry and Toxicology, Faculty of Pharmacy, University of Valencia, Av. Vicent Andrés Estellés s/n, Burjassot, 46100 València, Spain

**Keywords:** coffee, anti-inflammatory activity, ROS, skatole, chlorogenic acids, polyphenols

## Abstract

Coffee consumption positively influences colon health. Conversely, high levels of tryptophan metabolites such as skatole released from intestinal putrefactive fermentation in the presence of excessive dietary animal protein intake, and gut microbiota alterations, may have several adverse effects, including the development of colorectal cancer. Therefore, this study aimed to elucidate the potential protective effects of coffee in the presence of different skatole levels. The results showed that skatole exposure induced reduced cell viability and oxidative stress in the HT-29 human colon cancer cell line. However, co-treatment of cells with skatole and coffee samples was able to reduce ROS production (up to 45% for espresso) compared to cells not treated with coffee. Real-time PCR analysis highlighted that treating HT-29 cells with skatole increased the levels of inflammatory cytokines and chemokines TNF-α, IL-1β, IL-8, and IL12, whereas exposure to coffee extracts in cells that were pretreated with skatole showed anti-inflammatory effects with decreased levels of these cytokines. These findings demonstrate that coffee may counteract the adverse effects of putrefactive compounds by modulating oxidative stress and exerting anti-inflammatory activity in colonocytes, thus suggesting that coffee intake could improve health conditions in the presence of altered intestinal microbiota metabolism.

## 1. Introduction

Coffee is a highly popular beverage consumed throughout the centuries around the world [1], and a rising volume of scientific data is supporting the apparent health benefits of regular coffee consumption [2,3]. Coffee contains a large variety of relevant bioactive molecules, such as alkaloids, fibers, melanoidins, and polyphenols, which appear to be responsible for its pharmacological actions [4,5]. Several studies have reported that the positive effects ascribed to coffee intake include protection against type 2 diabetes, cardiovascular disease, and obesity, and as well as a wide range of cancer types [6].

It is well established that natural antioxidants have a protective effect against several lifestyle-related diseases, including colon cancer, by efficiently reducing reactive oxygen species (ROS) formation and preventing colonic inflammation [7,8,9]. Inflammation is recognized as an immune response that is triggered by microbial infection or tissue damage in humans [10]. As a result of this response, several pro-inflammatory cytokines are released, including interleukin-12 (IL-12), IL-6, IL-1β, chemokine IL-8, tumor necrosis factor-alpha (TNF-α), and interferon gamma (IFN-γ) [11]. Moreover, a variety of anti-inflammatory cytokines are generated, such as IL-10 and IL-4 [12]. Furthermore, the increased formation of ROS may prompt tissue damage and inflammation of intestinal mucosa, resulting in an higher risk of cancer development [13,14].

An ever-growing number of scientific surveys evidence the pivotal role of the intestinal microbiota in many human functions, including nutrition digestion and absorption, production of vitamins, and the modulation of innate immunity [15,16,17,18,19]. Intestinal microbiota converts dietary components, releasing a wide range of compounds that can have either positive or adverse effects on the health of the host [20]. A large body of evidence indicates that consumption of processed and red meat, and a high-fat diet, can be associated with altered microflora fermentation processes and enhanced production of putrefactive metabolites [21,22]. These compounds include indolic compounds such as 3-indoxyl sulfate and 3-methyl-indole (also known as skatole) and originate from the microbial catabolism of tryptophan [23,24]. Several investigations have found that high levels of these tryptophan metabolites may lead to unhealthy outcomes, including hepatic coma [25], rheumatoid arthritis [26], pulmonary edema [27], and schizophrenia [28,29]. In this context, although much remains to be understood regarding the molecular effects of these compounds on gut homeostasis, skatole has been linked to the progression and pathogenesis of inflammatory bowel disease (IBD), and to the development of colorectal cancer [30,31,32]. As recently reported in the literature, high skatole levels may exert pro-inflammatory properties [33,34]. Furthermore, Karlin et al. [35] reported that subjects with colorectal cancer excreted a higher concentration of fecal skatole than subjects without cancer (*p* < 0.01), highlighting a relationship between colorectal cancer and the level of fecal skatole.

Coffee consumption has been linked to a lower incidence of colorectal cancer, possibly due to the potent anti-inflammatory and antioxidant capacity of the active compounds in coffee [36]. Scientific evidence has been reported that shows active compounds in coffee, mainly caffeoylquinic acid (CQA) and dicaffeoylquinic acid (diCQA), have a protective action against ROS formation and DNA damage in human cell models [37,38]. Researchers conducting experimental studies in mice have reported that coffee reduced TNF-α in adipose tissue and decreased the expression of IL-1B and IL-6 in serum [39,40]. In addition, coffee compounds affect the homeostatic balance of intestinal microbiota, suppressing the bacterial N-acetyltransferase activity [41,42]. Several in vivo and in vitro studies have highlighted that coffee consumption is able to increase the *Bifidobacterium* spp. concentration in the colonic stage, as opposed to that of *Bacteroides* and *Clostridium* spp., with well-known positive effects, implying that coffee intake may have a prebiotic impact [43,44,45].

In a recent study, we identified and quantified the main bioactive compounds, including polyphenols, present in three different kinds of coffee brews using an UHPLC-Q-Orbitrap HRMS [46]. In our previous investigation, performed in the human colorectal adenocarcinoma HT-29 cell line [47], we reported that coffee brews reduced the intracellular ROS levels, probably due to their high content of bioactive compounds such as polyphenols. Moreover, after simulated gastrointestinal (GI) digestion, coffee showed fewer cytotoxic effects in the MTT test and a greater reduction in IL-6 levels than in the undigested samples.

Although the effects of coffee consumption on intestinal microbiota modulation have been partly clarified, a more complete understanding is required to elucidate its protection against colon cancer in presence of altered microbial fermentation processes. Hence, the current study aimed to explore the potential anti-inflammatory and antioxidant properties of different coffee samples in HT-29 cells exposed to various concentrations of skatole and coffee extracts.

## 2. Materials and Methods

### 2.1. Sampling

Three different kinds of coffee brews were investigated in the present article: instant coffee (n = 10), espresso (n = 10), and Americano coffee brews (n = 10). Instant coffee powder/granule samples and medium-roasted coffee beans (*Coffea arabica* L.) were obtained from local Italian markets. The coffee brews were prepared as described in our previous work [46] and reported in the Appendix A.

### 2.2. In Vitro Gastrointestinal Digestion

In vitro GI digestion was performed on the three coffee brew samples under investigation to simulate the effects of human digestion using the protocol proposed by the INFOGEST network, as used in our previous article [46]. The protocol employed is reported in the Appendix A.

### 2.3. Cell Culture

The human colorectal cancer cell line HT-29 from American Type Culture Collection (ATCC, Manassas, VA, USA) was maintained in high glucose RPMI 1640 medium, supplemented with 10% heat-inactivated fetal bovine serum plus 4 mM glutamine, at 37 °C in a humidified atmosphere containing 5% CO_2_. Cells were passaged by trypsinization when reaching 70–80% confluence (all reagents were from Sigma-Aldrich, Saint Louis, MO, USA). To avoid mycoplasma contamination, cells were routinely checked with a PCR Mycoplasma Test Kit (AppliChem A3744, Darmstadt, Germany).

### 2.4. Cell Treatment

HT-29 cells were treated either with coffee samples (espresso, Americano, and instant coffee brew) prepared in the cell culture medium at 0.250 and 0.500 mg/mL as previously reported [47], or with different concentrations of skatole (#M51458-5G/Sigma-Aldrich, Saint Louis, MO, USA) prepared from a 1 M DMSO stock solution in accordance with previous studies [48]. Furthermore, co-treatment experiments were performed in HT-29 cells treated with the assayed coffee samples and different concentrations of skatole for 24 h. Appropriate control cell cultures treated with the same amount of DMSO were included in each experiment and maintained at a final concentration of DMSO (*v*/*v*) of under 0.5% in the mock control. Cells were subsequently seeded to perform a cell viability assay, evaluate the intracellular ROS level, and to extract total mRNA for real-time PCR analysis.

### 2.5. Analysis of Cell Viability

Cell viability was measured using a thiazolyl blue tetrazolium bromide (MTT) colorimetric method (Roche, Mannheim, Germany) following the procedure described by Riccio et al. [49]. In brief, HT-29 cells were seeded onto 96-well plates at a density of 5.5 × 10^4^ cells/mL in 100 µL cell suspension per well. After 24 h, cells were treated with different skatole concentrations (250, 500, 750, 1000 µM) or with coffee samples (0.250 and 0.500 mg/mL) for 24 h. Then, 10 µL of MTT labeling reagent (Cell Proliferation Kit I; Roche, Mannheim, Germany) was added to the cell culture. After 4 h of incubation at 37 °C to dissolve the MTT insoluble formazan crystals, 100 µL of detergent solubilization buffer 1× (10% SDS in 0.01M HCl) was added to each well, according to the manufacturer’s instructions. The absorbance was read at 570/690 nm using a Synergy H1 Hybrid Multi-Mode Microplate Reader (BioTek, Winooski, VT, USA). The cell viability was calculated as a percentage as follows: (absorbance of the experimental group/absorbance of the control group) × 100.

### 2.6. Assessment of Intracellular ROS Production

The generation of intracellular ROS was measured using a spectrofluorometric test using an H_2_DCF-DA (2′7′-dichlorodihydrofluorescein diacetate) fluorescent probe [50]. HT-29 cells were plated onto 96-well black plates at a density of 5.5 × 10^4^ cells/mL in 100 µL cell suspension per well. Cells were treated with assayed coffee samples (0.250 and 0.500 mg/mL) or skatole (250 and 500 µM) for 24 h. In the positive control, ROS were generated by incubating HT-29 cells with 100 µM of hydrogen peroxide (H_2_O_2_), followed by H_2_DCF-DA incubation as previously described [47]. To evaluate the intracellular ROS levels under challenging conditions, cells were pretreated with 250 or 500 µM of skatole for 6 h and then treated with 0.250 or 0.500 mg/mL of coffee samples and skatole (250 or 500 µM) for an additional 18 h. After treatment, Dulbecco’s phosphate buffered saline (DPBS) was used to wash the cells twice. The cells were then exposed to 10 µM of H2DCF-DA diluted in Hank’s Balanced Salt Solution (HBSS) for 20 min in the dark at 37 °C. The extracellular dye was then removed from the cells by two washes with 1 × DPBS. The fluorescence intensity was detected using a Synergy H1 Hybrid Multi-Mode Microplate Reader (BioTek) at excitation/emission wavelengths of 485/538 nm. The percentage of intracellular ROS was calculated as follows: (fluorescence intensity of the experimental group/fluorescence intensity of the control group) × 100.

### 2.7. RNA Extraction

HT-29 cells were plated onto 6-well plates at a density of 1.5 × 10^5^ cells/mL. Cells were treated with skatole at a concentration of 250 µM for 24 h. In the positive control, inflammation response was stimulated by incubating the HT-29 cells with 10 ng/mL of LPS (Sigma Aldrich) [51,52]. In order to test inflammation in challenging conditions, cells were pretreated with skatole (250 µM) for 6 h, and then treated in combination with the assayed coffee samples (0.250 mg/mL) plus skatole (250 µM) for an additional 18 h. After treatment, cells were trypsinized and harvested to perform total RNA extraction. Total RNAs were extracted with QIAzol reagent (Qiagen, GmbH, Hilden, Germany) according to the manufacturer’s protocol. RNA was checked for purity and stability using gel electrophoresis and UV spectrometry. Absorption at 260 and 280 nm was measured, and RNA quantity was calculated.

### 2.8. Real-Time PCR Analysis

Real-time PCR analysis was performed with 1 µg of RNA reverse-transcribed using the iScript Reverse Transcription Supermix for RT-qPCR (Bio-Rad, Berkeley, CA, USA) in a final volume of 20 µL, according to the manufacturer’s instructions. This mixture was incubated at 42 °C for 3 min, and then at 95 °C for 3 min, and subsequently used for real-time RT-PCR procedures on a CFX96 Real-Time System (Bio-Rad Laboratories, Hercules, CA, USA).

Primers for quantitative real-time PCR analysis are reported in Table 1 [53]. β-actin mRNA was used as an endogenous control. Each real-time PCR experiment was performed in triplicate in a 20 μL reaction mix containing 10 μL of 2× SsoAdvanced Universal SYBR Green Supermix (Bio-Rad Laboratories), 0.38 μL of a 20 μM primer mix, and 6.6 μL of cDNA (1/2 volume of RT-PCR product). The cycling conditions were set up as follows: initial denaturation step at 98 °C for 30 s, followed by 40 cycles (95 °C for 15 s, 60 °C for 30 s). A calibration curve was calculated to assess the efficiency of the PCR, as previously reported [54]. Real-time PCR reactions were performed using the CFX Opus 96 Real-Time PCR System (Bio-Rad Laboratories) and CT values were obtained from automated threshold analysis. Data were analyzed using CFX Manager 3.0 software (Bio-Rad Laboratories GmbH, Munich, Germany) according to the manufacturer’s specifications, and a relative quantification of gene expression was determined using the ΔΔCT method.

### 2.9. Statistical Analysis

The experiments were carried out in triplicate and the results are provided as the mean ± standard deviation (SD). A one-way analysis of variance (ANOVA) or two-way ANOVA test was used to determine the statistical differences between the control and treated cell groups. Where appropriate, Dunnett’s test and/or Student’s *t*-tests were performed. The *p*-values of 0.05, 0.01, and 0.001 (*, **, and ***, respectively) were used to determine the level of significance.

## 3. Results

### 3.1. Cell Viability in HT-29 Cells

The HT-29 cell viability after the exposure to the assayed coffee samples was investigated. Figure 1 shows the cell viability evaluated with an MTT assay after treatment with different coffee extracts (0.250 and 0.500 mg/mL). The results highlighted that the cells treated with the instant and Americano coffee samples at both assayed concentrations (0.250 and 0.500 mg/mL), showed a significant dose-dependent increase in cell viability (10% and 20%, respectively) compared to the mock control, whereas no significant differences were observed between those treated with espresso coffee and the mock control. To exclude effects mediated by the digestion fluid, the blank control resulting from in vitro digestion was analyzed by MTT assay and compared to untreated cells. The results (Figure 1A–C) indicated no significant differences between the blank control and untreated cells at 24 h of treatment. In addition, the cytotoxic effects of skatole treatment at different concentrations (from 250 to 1000 µM) on HT-29 cells was investigated by MTT assay after 24 h exposure (Figure 1D). As shown in Figure 1D, the colorimetric assay indicated no cytotoxicity effects in cells exposed for 24 h to skatole concentrations lower than 500 µM. However, at higher concentrations (750 µM and 1000 µM) a dose-dependent decrease was observed in cell viability, equal to 87% and 78% of that in untreated cells, respectively. Since exposure to the lower doses of coffee extracts (0.250 mg/mL) or skatole (250 and 500 µM) did not affect cell viability, these conditions were used for subsequent experiments.

### 3.2. Intracellular ROS Levels in HT-29 Cells

The effect of skatole and the three types of assayed coffee samples on intracellular ROS generation in HT-29 cells was examined after 24 h using a fluorometric test employing the specific dye H_2_DCF-DA. Cells treated only with H_2_O_2_ were used as a positive control. The intracellular ROS level results are shown in Figure 2. The fluorescence intensity (Figure 2A) decreased remarkably following coffee treatment at both 0.250 and 0.500 mg/mL concentrations, compared with the control group that received no coffee treatment. Specifically, in cells treated with coffee extracts at 0.250 and 0.500 mg/mL, the reduction in fluorescence intensity was 19% and 36.5% for instant coffee, 17.5% and 36% for Americano coffee, and 21.5% and 31.2% for espresso coffee, respectively. A blank control resulting from in vitro digestion was also examined and compared to the mock control to exclude side effects mediated by the digestion fluid, and it showed no significant differences compared to the control cells (Figure 2A). In addition, the possible pro-oxidant activity of skatole at non-toxic concentrations (250 and 500 µM) after 24 h of treatment was investigated (Figure 2B), showing significantly increased intracellular ROS levels in cells treated with skatole at 250 µM (13%) and 500 µM (8%) compared to the mock control (cells treated only with DMSO).

In order to examine the antioxidant capacity of the assayed coffee samples, we used an H_2_DCF-DA assay to evaluate the intracellular ROS levels in HT-29 cells after 24 h treatment with skatole (250 and 500 µM) and the respective coffee treatments at 0.250 mg/mL. The data obtained are shown in Figure 3. After stimulation with 250 µM of skatole (Figure 3A), fluorescence intensity significantly decreased for all assayed coffee samples, at a rate of 28% for instant coffee, 39% for Americano coffee, and 45% for espresso coffee. After stimulation with 500 µM of skatole (Figure 3B), the fluorescence intensity decreased for all assayed samples at the following rates: 11% for instant coffee, 30% for Americano coffee, and 39% for espresso coffee.

### 3.3. Anti-Inflammatory Effects of Coffee Extracts on Cytokine mRNA Expression Levels in HT-29 Cells

Real-time PCR analysis was performed to investigate the expression levels of mRNAs encoding the pro-inflammatory cytokines and chemokines TNF-α, IL-1β, IL-8, and IL12 in HT-29 cells under challenging conditions. In accordance with the literature [5,9], cells treated only with LPS were used as a positive control. Our data showed that cells treated with 250 µM of skatole for 24 h led to a significant up-regulation of TNF-α, IL-1β, IL-8, and IL-12 (1.61-, 3.07-, 1.92-, and 2.17-fold changes with respect to the mock control, respectively) (Figure 4). On the other hand, co-treatments with coffee extracts counterbalanced the pro-inflammatory effects mediated by skatole by down-modulating the expression of the analyzed cytokines to values almost comparable with the mock control. In particular, the mRNA expression levels of TNF-α (Figure 4A), IL-1β (Figure 4B), IL-8 (Figure 4C), and IL-12 (Figure 4D) were found to be significantly decreased when the cells were co-treated with espresso coffee (0.84-, 1.08-, 0.64-, and 0.55-fold changes with respect to the mock control, respectively). However, the levels of IL-8 (Figure 4C) and IL-12 (Figure 4D) were found to be markedly decreased in cells co-treated with instant coffee (1.13- and 1.19-fold changes with respect to the mock control, respectively), and IL-12 levels (Figure 4D) also decreased considerably upon co-treatment with Americano coffee (a 1.31-fold change with respect to the mock control).

## 4. Discussion

The main goal of the current study was to assess the potential protective effect of coffee against pro-inflammatory and pro-oxidant conditions triggered by putrefactive compounds released in the presence of an altered intestinal microbiota through the HT-29 human colon cancer cell line model. Due to its ability to express characteristics of mature intestinal cells, the human colon adenocarcinoma cell line HT-29 is effectively used not only to study the biology of human colon cancers, but is also attracting particular attention in studies on food digestion and bioavailability [55]. Notwithstanding the presence of several scientific investigations in the literature with regard to the positive effects of coffee intake on the prevention of colon cancer [56,57,58,59], there is still scarce knowledge concerning the potential anti-inflammatory and antioxidant properties of coffee in the presence of the putrefactive compounds that could be released in gut dysbiosis conditions.

In the present work, in order to mimic the effect of oral, gastric, and intestinal digestion, the INFOGEST protocol was followed [60]. This procedure is widely regarded as one of the most effective protocols to simulate the natural digestion process. In vitro intestinal models represent the gold standard in such investigations; in fact, they can rapidly provide useful information on the impact of food components on health status [61]. Despite the fecal inoculum method representing the most appropriate protocol to replicate in vitro colonic digestion, an increasing number of studies have reported that the combination of bacterial enzymes, such as Pronase E and Viscozyme L, represents a suitable alternative to reproducing intestinal fermentation [62,63,64,65,66,67].

In summary, we initially tested the cytotoxicity of skatole on HT-29 cells. The concentration range of skatole chosen to conduct the study ranged from 250 to 1000 µM based on previous studies reporting 1000 µM as the maximum skatole concentration found in human feces [48]. Our data revealed that HT-29 cell viability was affected in a concentration-dependent manner at the higher skatole concentrations tested (750 and 1000 µM), with a significant reduction in cell viability compared to control cells (87% and 78%, respectively). Kurata et al. [48] evaluated the effects of different skatole concentrations on cell viability using Caco-2 cells, another human colon cancer cell line, showing that skatole promoted apoptosis in these cells in a dose-dependent and time-dependent manner. Their in vivo studies demonstrated that urinary skatole levels in patients significantly decreased after the consumption of probiotic formulations containing *Bifidobacterium* spp. and *Lactobacillus* spp. [24].

On the other hand, no cytotoxicity effects were detected in HT-29 cell analysis for espresso coffee samples, but in contrast, a minimal increase in cell viability, up to 20%, was shown for instant and American coffee samples. Moreover, our data suggest that all the assayed coffee samples had an important effect in suppressing ROS formation in HT-29 cells, owing to their antioxidant activity. At the lowest assayed concentration (0.250 mg/mL), the espresso coffee sample appeared to exert a greater reduction in ROS levels than the other coffee brew samples. Moreover, the results highlighted that even at low concentrations (250 and 500 µM), skatole was able to induce an increase in intracellular ROS levels, in particular, at 250 µM of skatole the ROS levels increased by 13%, while at 500 µM of skatole the ROS levels increased by 8%, compared to a control. The data showed that skatole treatment represents a demanding setting for oxidative stress, which promotes raised intracellular ROS levels in HT-29 cells. Interestingly, data obtained in the present scientific study demonstrated that the treatment of HT-29 cells with different types of coffee samples in the presence of skatole significantly decreased the production of ROS compared to untreated cells. Among the different kinds of coffee brews tested, espresso showed the most effective antioxidant activity, reaching a 45% decrease in ROS compared with untreated cells.

In this study, the expression of genes linked to inflammation was also examined. The results showed that skatole exposure triggered an increased expression of pro-inflammatory cytokines and chemokines TNF-α, IL-1β, IL-8, and IL12 in HT-29 cells. These findings are consistent with the literature, which suggests that skatole may have pro-inflammatory effects and reports that this tryptophan metabolite may play a role in the development of colorectal cancer and in the progression and pathogenesis of IBD [68,69,70]. Moreover, our data clearly indicated that all three types coffee studied exhibit anti-inflammatory activity by decreasing the expression levels of cytokines in cells that were pretreated with skatole.

In this context, it has to be highlighted that espresso coffee showed both the highest antioxidant and anti-inflammatory properties among the coffee brews tested. Notably, a possible explanation for these findings could be related to one of our previous investigations, in which we reported that the espresso coffee sample shows a higher polyphenolic content than the other studied coffee samples, as analyzed using a UHPLC-Q-Orbitrap HRMS [46]. CGAs were the predominant polyphenols quantified in the coffee samples, notably the three CQA isomers, which accounted for 66% to 71% of the total polyphenols. As reported by scientific evidence [71,72], CGAs exert a strong antioxidant activity and inhibit the expression of inflammatory factors. Therefore, these results may partly explain the improved reduction in ROS levels and the enhanced anti-inflammatory activity found after treating HT-29 cells with espresso samples in the presence of skatole, indicating that the higher level of polyphenols detected in espresso coffee samples may play an important role in limiting the formation of ROS by exerting anti-inflammatory effects in the presence of putrefactive compounds.

## 5. Conclusions

In summary, our data indicate that direct treatment with skatole induced cytotoxicity in the HT-29 human colon cancer cell line in a concentration-dependent manner, resulting in a significant reduction in cell viability with respect to control cells. Moreover, the simultaneous treatment of HT-29 cells with coffee and skatole was able to decrease ROS production compared to control cells. Furthermore, HT-29 cells treated with skatole showed increased expression levels of the pro-inflammatory cytokines and chemokines TNF-α, IL-1β, IL-8, and IL12. Finally, our data demonstrate that all three types of coffee analyzed exhibited anti-inflammatory activity by hampering the up-regulation of pro-inflammatory cytokines induced by skatole exposure. These findings highlight that coffee could exert anti-inflammatory activity and mitigate oxidative stress in the presence of high levels of putrefactive compounds, suggesting that coffee consumption may improve health conditions by modulating the risk of colorectal inflammation.

## Figures and Tables

**Figure 1 antioxidants-11-02458-f001:**
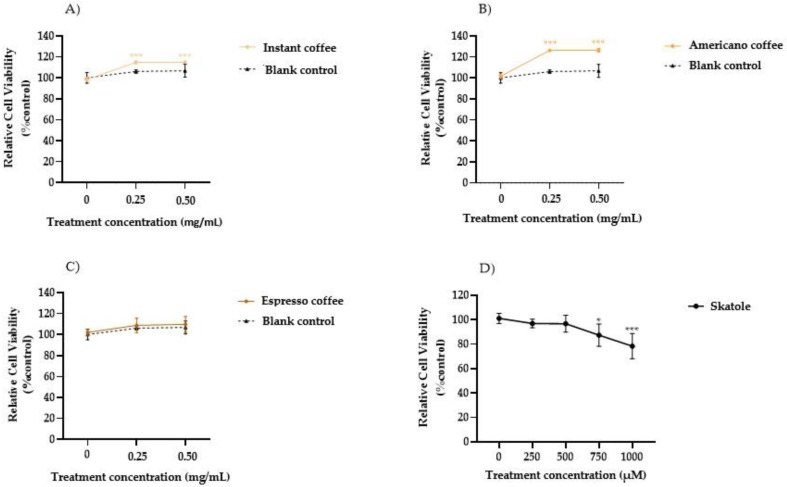
Evaluation of cell viability in HT-29 cells. The effect of treatment with different types of coffee extract: instant (**A**), Americano (**B**), and espresso (**C**), at 0.250 and 0.500 mg/mL on cell viability was evaluated using the MTT assay after 24 h with respect to the mock control. The MTT test was used to determine the impact of skatole treatment (**D**) on cell viability after 24 h at the concentrations of 250, 500, 750, and 1000 µM compared with control cells (treated only with DMSO). The graph represented the mean and SD of three separate experiments. * *p*-value ≤ 0.05 and *** *p*-value ≤ 0.001 compared to the control group (calculated as fold-change relative to control cells, arbitrarily set at 100%).

**Figure 2 antioxidants-11-02458-f002:**
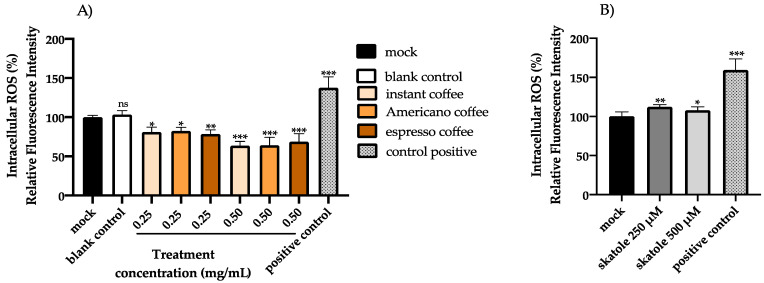
Evaluation of intracellular ROS level in HT-29 cells. The effect on the generation of intracellular ROS levels after treatment with a blank control (resulting from in vitro digestion) and the different types of coffee extract (instant, Americano, and espresso) at 0.250 and 0.500 mg/mL was evaluated using the H_2_DCF-DA assay after 24 h of treatment, and compared with the mock control (untreated cells) (**A**). The effect of skatole treatment (**B**) on the production of intracellular ROS was estimated by fluorometric assay after 24 h at the concentrations of 250 and 500 µM and compared to the control cells (treated with DMSO only). Cells treated only with H_2_O_2_ were used as a positive control. The graphs represent the mean and SD of three separate experiments. * *p*-value ≤ 0.05, ** *p*-value ≤ 0.01 and *** *p*-value ≤ 0.001 compared to the control group (calculated as fold-change relative to control cells, arbitrarily set at 100%).

**Figure 3 antioxidants-11-02458-f003:**
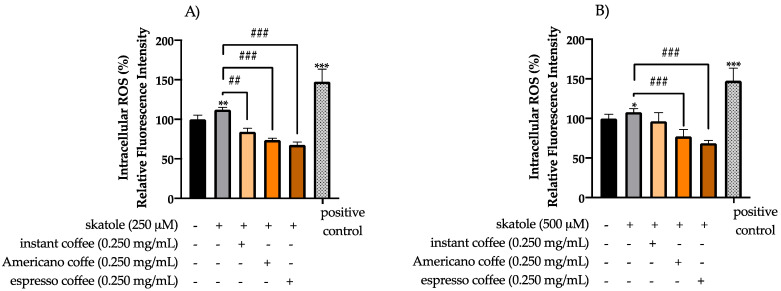
Evaluation of intracellular ROS levels under challenging conditions in HT-29 cells. The effect of treatment with different types of coffee extract (instant, Americano, and espresso) at 0.250 mg/mL after skatole treatment of 250 µM (**A**) and 500 µM (**B**) on the production of intracellular ROS levels was determined by H_2_DCF-DA assay after 24 h of treatment, and compared to the mock control. Cells treated only with H_2_O_2_ were used as a positive control. The graphs represent the mean and SD of three separate experiments. * *p*-value ≤ 0.05, ** *p*-value ≤ 0.01 and *** *p*-value ≤ 0.001 compared to untreated control (calculated as fold-change relative to control cells, arbitrarily set at 100%). ## *p*-value ≤ 0.01 and ### *p*-value ≤ 0.001 skatole versus coffee treatment.

**Figure 4 antioxidants-11-02458-f004:**
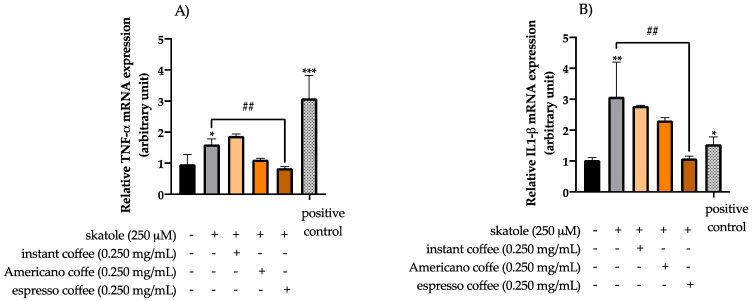
Evaluation of mRNA expression levels in HT-29 cells. The effect of treatment with the different types of coffee extract (instant, Americano and espresso) at 0.250 mg/mL after stimulation with 250 µM skatole on the expression level of TNF-α (**A**), IL-1β (**B**), IL-8 (**C**), and IL12 (**D**) was performed using real-time PCR analysis after 24 h of treatment, and compared to the control group. Cells treated only with LPS were used as a positive control. The graph represented the mean and SD of three separate experiments. * *p*-value ≤ 0.05, ** *p*-value ≤ 0.01 and *** *p*-value ≤ 0.001 compared to untreated control. # *p*-value ≤ 0.05, ## *p*-value ≤ 0.01 and ### *p*-value ≤ 0.001 skatole versus coffee treatment.

**Table 1 antioxidants-11-02458-t001:** Primer sequences used for quantitative Real-time PCR analysis.

Transcript	Primer	Sequence 5′-3′	Amplicon Size (bp)
TNF-α	For	AGCCCATGTTGTAGCAAACC	134
	Rev	TGAGGTACAGGCCCTCTGAT	
IL-1β	For	CATGGGATAACGAGGCTTATG	149
	Rev	CCACTTGTTGCTCCATATCC	
IL-8	For	TGGCTCTCTTGGCAGCCTTC	238
	Rev	TGCACCCAGTTTTCCTTGGG	
IL-12	For	TTCACCACTCCCAAAACCTGC	225
	Rev	GAGGCCAGGCAACTCCCATTA	
β-actin	For	CGACAGGATGCAGAAGGAGA	160
	Rev	CGTCATACTCCTGCTTGCTG	

## Data Availability

Data is contained within the article and Appendix A.

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
