# Peer review of "Effect of Different Coffee Brews on Tryptophan Metabolite-Induced Cytotoxicity in HT-29 Human Colon Cancer Cells"

_antioxidants, 2022, doi:10.3390/antiox11122458_

Round 1

Reviewer 1 Report

In the manuscript entitled:” Effect of Different Coffee Brews on Tryptophan Metabo-2 lite-Induced Cytotoxicity in HT-29 Human Colon Cancer Cells,” the Authors used HT-29, a human colorectal adenocarcinoma cell line widely used to study drug delivery and nutritional effects. HT-29s are also an excellent model for studying the anti-cancer effects of various compounds. However, the proposed model suggests that the different coffee blends are protective against cancer cells (HT-29) which is counterintuitive given that usually, the aim is to kill cancer cells, not to protect them. Therefore, the protective effects must be studied in a model of Normal human intestinal epithelial cells (i.e., HIEC-6).

Author Response

Reviewer 1

In the manuscript entitled:” Effect of Different Coffee Brews on Tryptophan Metabo-2 lite-Induced Cytotoxicity in HT-29 Human Colon Cancer Cells,” the Authors used HT-29, a human colorectal adenocarcinoma cell line widely used to study drug delivery and nutritional effects. HT-29s are also an excellent model for studying the anti-cancer effects of various compounds. However, the proposed model suggests that the different coffee blends are protective against cancer cells (HT-29) which is counterintuitive given that usually, the aim is to kill cancer cells, not to protect them. Therefore, the protective effects must be studied in a model of Normal human intestinal epithelial cells (i.e., HIEC-6).

As well-reviewed by Daniel Martínez-Maqueda et al, (The Impact of Food Bioactives on Health: in vitro and ex vivo models, doi: 10.1007/978-3-319-16104-4) the human colon adenocarcinoma cell line HT-29 is not only used to study the biology of human colon cancers, but it is receiving special interest in studies focused on food digestion and bioavailability due to the ability to express characteristics of mature intestinal cells. Moreover, Daniel Martínez-Maqueda et al., concluded "Altogether, they are a complementary tool to the in vivo and ex vivo strategies to study food digestion and the effect of food components on the gut".

An ever-expanding amount of scientific studies are investigating the protective and potential anticancer effects of food compounds using these cell lines, and their gastrointestinal digests. (doi:10.1016/j.jff.2013.01.036; doi: 10.1111/1750-3841.14102; doi: 10.18632/oncotarget.20216).

Based on these observations, the authors believe that HT-29 cells represent a suitable model for this study since the antioxidant and anti-inflammatory effects detected in these cells are to be considered as protective against the transformation of normal intestinal epithelial cells into cancer cells without any implication for anti-cancer therapies. However, as suggested by Reviewer 1, the authors added some information in the manuscript to explain the choice of this cell model.

Reviewer 2 Report

The manuscript “Effect of Different Coffee Brews on Tryptophan Metabolite-Induced Cytotoxicity in HT-29 Human Colon Cancer Cells” aims to elucidate the potential protective effects of coffee in the presence of different skatole levels.

In my opinion it is interesting as a study, but some clarifications and corrections are needed before publication:

-Interleukin 8 is improperly defined as a cytokine throughout the text, it would be better to define it as a chemokine.

-Names of bacteria mentioned in the text should be checked and formatted in italics

-In lines 40-42 I recommend reading and adding the quote to the paper DOI: 10.1016/j.biotechadv.2018.11.011

- In the results (line 208) why are HT-25s mentioned? Didn't we talk about the HT-29?

-In the cell viability results, I think it is appropriate to show the data for the entire range used and not just for the concentrations used for subsequent experiments.

- In the discussion a few more sentences should be added regarding the results obtained with other natural compounds in other studies, so as to have a more constructive overall picture.

-Check the formatting of all references (eg the use of italics), because it does not follow the guide to authors and add the doi for all references used.

Author Response

Reviewer 2

The manuscript “Effect of Different Coffee Brews on Tryptophan Metabolite-Induced Cytotoxicity in HT-29 Human Colon Cancer Cells” aims to elucidate the potential protective effects of coffee in the presence of different skatole levels.

In my opinion it is interesting as a study, but some clarifications and corrections are needed before publication:

-Interleukin 8 is improperly defined as a cytokine throughout the text, it would be better to define it as a chemokine.

As suggested by Reviewer 2, the authors defined Interleukin 8 as a chemokine

-Names of bacteria mentioned in the text should be checked and formatted in italics

As suggested by Reviewer 2, the authors corrected the names of bacteria mentioned in the text.

-In lines 40-42 I recommend reading and adding the quote to the paper DOI: 10.1016/j.biotechadv.2018.11.011

As suggested by Reviewer 2, the authors added the reference in the manuscript.

 - In the results (line 208) why are HT-25s mentioned? Didn't we talk about the HT-29?

As suggested by Reviewer 2, the authors corrected the typo

-In the cell viability results, I think it is appropriate to show the data for the entire range used and not just for the concentrations used for subsequent experiments.

The authors appreciate the valuable advice of Reviewer 2 and understand the rationale of Reviewer 2 for recommending showing the data for the entire range used. However, to help the reader understand the experimental design, the authors believe it is more appropriate to present the data in their current format.

- In the discussion a few more sentences should be added regarding the results obtained with other natural compounds in other studies, so as to have a more constructive overall picture.

As suggested by Reviewer 2, the authors added more information regarding the results obtained with other natural compounds. Until now, the only natural compounds investigated in this matter are probiotics.

-Check the formatting of all references (eg the use of italics), because it does not follow the guide to authors and add the doi for all references used.

As suggested by Reviewer 2, the authors checked the formatting of all references. We would like to inform the Reviewer 2 that MDPI automatically adds DOIs to references.

The authors thank the Reviewers for evaluating our manuscript.

Reviewer 3 Report

The manuscript titled "Effect of Different Coffee Brews on Tryptophan Metabolite-Induced Cytotoxicity in HT-29 Human Colon Cancer Cells" deas with the possibilites of putrefative molecules to demage gut microbiome and cells leading to colon cancer and coffee brews benefit effect. 

The manuscript has some good aspects, but the major problem in these experiments is samples of coffee brews. There are some data from the previous author's paper and polyfenol profies, but there are only one example of each coffee brew bought in market. Let's all of us buy the comercial product and say these samples are for research. At least there should be 3 examples of each brew to be possible to conclude such a results. Otherwise, it is speculation. 

The figures are difficult to understand with all those plus and minuses

There are a lot of grammar mistakes, and too much crutch word like moreover, meanwhile, however. fOR EXAMPLE IN LINE 90 THERE IS however, although.......

ther is no word like COLONIC CANCER but COLON CANCER. 

Author Response

Reviewer 3

The manuscript titled "Effect of Different Coffee Brews on Tryptophan Metabolite-Induced Cytotoxicity in HT-29 Human Colon Cancer Cells" deas with the possibilites of putrefative molecules to demage gut microbiome and cells leading to colon cancer and coffee brews benefit effect. 

The manuscript has some good aspects, but the major problem in these experiments is samples of coffee brews. There are some data from the previous author's paper and polyfenol profies, but there are only one example of each coffee brew bought in market. Let's all of us buy the comercial product and say these samples are for research. At least there should be 3 examples of each brew to be possible to conclude such a results. Otherwise, it is speculation. 

As reported in our previous paper, the number of samples analyzed for each type of coffee was n=10. The authors report the material section of our previous article: "Espresso (n = 10) and Americano samples (n = 10) were delivered as vacuum-packed, medium-roasted coffee beans, whereas spray-dried, instant coffee samples (n = 10) were obtained in powder/granule form." As suggested by Reviewer 3, the authors explained better the amount of the sample size.

The figures are difficult to understand with all those plus and minuses

The authors agree with this observation. However, displaying the data in their current form represents a simple way that allows the reader to visualize the whole of the results obtained.

There are a lot of grammar mistakes, and too much crutch word like moreover, meanwhile, however. fOR EXAMPLE IN LINE 90 THERE IS however, although.......

As suggested by Reviewer 3, the authors checked the grammar mistakes in the manuscript.

ther is no word like COLONIC CANCER but COLON CANCER. 

The authors agree with this observation

The authors thank the Reviewers for evaluating our manuscript.

Round 2

Reviewer 1 Report

Consistent with the review of Daniel Martínez-Maqueda et al., mentioned by the Authors, the human colon adenocarcinoma cell line HT-29 is not only used to study the biology of colon tumors. “They are a complementary tool to the in vivo and ex vivo strategies to study food digestion and the effect of food components on the gut.

My concern is about using only this cell line to study the anti-inflammatory and antioxidant properties of various coffee samples in cells exposed to different concentrations of skatole and coffee extracts when as well described in the review mentioned above, the “HT29 secreted pro-inflammatory cytokines, such as, tumor necrosis factor (TNF) α and interleukins (IL) 1β and IL 6; growth factors such as platelet-derived growth factor AA and transforming growth factors (TGF) α and β; chemokines such as fractalkine, IL-8, monocyte chemoattractant protein-1 and interferon-γ-induced protein 10; pro-angiogenic factors such as IL-15 and vascular endothelial growth factor; and immune-modulatory cytokines such as granulocyte colony-stimulating factor, granulocyte macrophage colony-stimulating factor and IL-3.”

Author Response

Reviewer 1

Consistent with the review of Daniel Martínez-Maqueda et al., mentioned by the Authors, the human colon adenocarcinoma cell line HT-29 is not only used to study the biology of colon tumors. “They are a complementary tool to the in vivo and ex vivo strategies to study food digestion and the effect of food components on the gut.

My concern is about using only this cell line to study the anti-inflammatory and antioxidant properties of various coffee samples in cells exposed to different concentrations of skatole and coffee extracts when as well described in the review mentioned above, the “HT29 secreted pro-inflammatory cytokines, such as, tumor necrosis factor (TNF) α and interleukins (IL) 1β and IL 6; growth factors such as platelet-derived growth factor AA and transforming growth factors (TGF) α and β; chemokines such as fractalkine, IL-8, monocyte chemoattractant protein-1 and interferon-γ-induced protein 10; pro-angiogenic factors such as IL-15 and vascular endothelial growth factor; and immune-modulatory cytokines such as granulocyte colony-stimulating factor, granulocyte macrophage colony-stimulating factor and IL-3.”

The authors understand the observations of Reviewer 1, however, the authors prove that the HT-29 is not only used to study the biology of colon tumors, but they are also a complementary tool to the in vivo and ex vivo strategies to study food digestion and the effect of food components on the gut. In this line, a huge literature proposed the use of HT-29 cells as a suitable model for this type of study since the antioxidant and anti-inflammatory effects detected in these cells are to be considered as protective against the transformation of normal intestinal epithelial cells into cancer cells.

In addition, as reported (2012, DOI: https://doi.org/10.1124/dmd.111.042465), " Bourgine et al. compared the expression of 377 genes in HT29 and other intestinal cell lines used as in vitro models of the epithelium with the corresponding tissue biopsy, and the results showed that differentiated HT29 cells and human colonic tissues are not significantly different". Moreover, "the use of HT29 as in vitro model of intestinal cells has some advantages (Zweibaum et al., 2011, DOI:https://doi.org/10.1002/cphy.cp060407). This cell line in its differentiated phenotype is similar to small intestine enterocytes with respect to their structure, the presence of brush border-associated hydrolases, and the time course of the differentiation process which is also comparable to that found in the small intestine. In addition, the amount of villin expressed in differentiated HT29 cells is close to the value observed for normal freshly prepared colonocytes". Furthermore, "It has been postulated that these cells are close to human fetal colonic cells because of the type of hydrolases present and the intracellular concentration of glycogen accumulated (Hekmati et al. 1990, DOI:10.1016/0922-3371(90)90133-H)". In addition, "Regarding the expression of cell surface receptors, in general, the receptors found in this cell line have their equivalent in normal intestinal cells, except for the receptor to neurotensin".

Finally, the authors added some discussion in the manuscript to explain the choice of this cell model.

The authors thank Reviewer 1 for evaluating our manuscript.

Reviewer 3 Report

nothing to add